

**Ensemble cloud-resolving modelling of a historic back-building mesoscale**
**convective system over Liguria: The San Fruttuoso case of 1915**
Antonio Parodi[1], Luca Ferraris[1,2], William Gallus[3], Maurizio Maugeri[4], Luca Molini[1], Franco
Siccardi[1], and Giorgio Boni[1,2]

7       1- CIMA Research Foundation, Savona, Italy
8       2- Dipartimento di Informatica, Bioingegneria, Robotica e Ingegneria dei Sistemi,
9          University of Genoa, 16145 Genoa, Italy
3- Department of Geological and Atmospheric Sciences, Iowa State University, Ames, Iowa
4- Università degli Studi di Milano, Dipartimento di Fisica, Milan, Italy
**Abstract**
Highly localized and persistent back-building mesoscale convective systems represent
one of the most dangerous flash-flood producing storms in the north-western
Mediterranean area. Substantial warming of the Mediterranean Sea in recent decades
raises concerns over possible increases in frequency or intensity of these types of
events as increased atmospheric temperatures generally support increases in water
vapor content. However, analyses of the historical record do not provide a univocal
answer, but these are likely affected by a lack of detailed observations for older
events.
In the present study, 20th Century Reanalysis Project initial and boundary condition
data in ensemble mode are used to address the feasibility of performing cloud-
resolving simulations with 1 km horizontal grid spacing of a historic extreme event
that occurred over Liguria: The San Fruttuoso case of 1915. The proposed approach
focuses on the ensemble Weather Research and Forecasting (WRF) model runs that
show strong convergence over the Liguria sea, as these runs are the ones most likely
to best simulate the event. It is found that these WRF runs generally do show wind
and precipitation fields that are consistent with the occurrence of highly localized and
persistent back-building mesoscale convective systems, although precipitation peak
amounts are underestimated. Systematic small north-westward position errors with
regard to the heaviest rain and strongest convergence areas imply that the Reanalysis
members may not be adequately representing the amount of cool air over the Po Plain
outflowing into the Liguria Sea through the Apennines gap. Regarding the role of
historical data sources, this study shows that in addition to Reanalysis products,
unconventional data, such as historical meteorological bulletins newspapers and even
photographs can be very valuable sources of knowledge in the reconstruction of past
extreme events.



## 1. Introduction

Flash floods are phenomena very common to most Mediterranean coastal cities, accountable for millions of euros of damage and tens to hundreds of victims every year (Gaume et al. 2009). The north-western Mediterranean area is affected by such events in a period usually spanning from late summer (the end of August) to late fall (early December): in this period, the warm waters of the sea, in combination with large-scale meteorological systems coming from the Atlantic Ocean, provide a huge amount of energy, namely latent and sensible heat fluxes, to the atmosphere (Boni et al. 2006, Reale et al. 2011, Pinto et al. 2013). Heavy precipitation is then triggered by the typically very steep topography of the coasts: it is frequent to observe the monthly average rainfall to fall intensely in just a few hours and/or a significant fraction (up to 30-40%) of the yearly average in one day (Parodi et al 2012, Fiori et al. 2014). Obviously, the losses experienced in terms of human lives and economic damage in these very densely populated areas are often dramatic.

Among the flash flood producing storms in the Mediterranean area, a prominent feature is the highly localized and persistent back-building of mesoscale convective systems (MCSs, Schumacher and Johnson 2005, Duffourg et al. 2015, Violante et al. 2016). Such a scenario has been observed often in the last decade, when Liguria (NW Italy) and Southern France have been repeatedly hit by severe floods: 2010 Varazze and Sestri Ponente, 2011 Cinqueterre and Genoa, 2012 Marseille and Isle du Levant, 2014 Genoa and Chiavari, 2015 Nice. As shown in several recent works (Parodi et al. 2012, Rebora et al. 2013, Fiori et al. 2014, Duffourg et al 2015, Silvestro et al. 2015, Cassola et al. 2016, Silvestro et al. 2016), convective cells, embedded in such MCSs, are generated on the sea by the convergence of a warm and moist south-easterly flow and a northerly much colder and drier one. These structures are then advected to the land where the combined action of the aforementioned currents and the topography force them to persist for several hours over a very localized area (e.g. about 100 km$^2$).

Many flood frequency studies have been carried out, focusing on rainfall regimes and Mediterranean flood seasonality and type (Barriendos et al. 2003, Llasat et al. 2005, Barriendos et al. 2006, Boni et al. 2006, Pinto et al. 2013, Llasat et al. 2014, Toreti et al. 2015). Due to the exploitation of both documentary sources and early measurements, these analyses have been able to go back several centuries, however, their results have been mostly inconclusive regarding changes in frequency of occurrence. Well-defined trends have not been found as usually flood frequency oscillates from period to period with no significant growth, not even in the most recent decades, regardless of the event's duration (a few hours to days).

The same result applies to precipitation extremes and their possible changes over the Mediterranean area in recent decades, studied by several authors, either by empirical or (mainly at-site) extreme value theory approaches (see e.g. Brunetti et al., 2001, 2004, Alpert et al., 2002, Kostopoulou and Jones, 2005, Moberg et al., 2006, Brunet et al., 2007, Kioutsioukis et al., 2010, Rodrigo, 2010, Toreti et al., 2010, van den Besselaar et al., 2013). The temporal tendencies are not fully coherent throughout the region (Ulbrich et al., 2012) and rather conditioned by the specific site, the approach used and the period examined (Brugnara et al., 2012, Brunetti et al., 2012, Maugeri et al., 2015). On the contrary, an increase in precipitation extremes over the Mediterranean area is generally indicated by climate model scenarios (Alpert et al., 2002, Giorgio and Lionello, 2008, Trenberth, 2011).



It is therefore still an open debate whether the frequency of these phenomena is
really increasing or if it is merely the perception of both the general public and
scientific community. The latter hypothesis is supported by the fact that in the last
10-20 years the observational capabilities have substantially increased. For example,
in Italy alone, the remotely automated weather station network has grown to 5000
stations offering an average density of about 1/75 station/km$^2$ with a 1 to 10-minute
sampling rate. At the same time, the national weather radar network reached a fully
operational coverage allowing for direct evaluation of the space-time structure of
precipitation (Rebora et al. 2012).
Other factors contributing to enhance the perception of an increasing frequency of
extreme precipitation and floods are that it has become much easier for weather-
related disasters to make it to the news (Pasquaré and Oppizzi 2012, Grasso and
Crisci 2016) and therefore to the general public, and that a rapidly growing population
and soil consumption increases the exposure of the population to such phenomena
(Ward et al. 2013, SOER 2015).
To better investigate whether extreme precipitation and flood frequency are really
increasing in the Mediterranean, it is important to improve the exploitation of the
information available from past meteorological data. A contribution to this
improvement may come from the development of methods that identify which
ensemble analyses from projects like the 20th Century Reanalysis Project are able to
produce precipitation fields that are reasonably intense and capable of causing
extreme floods.
This paper focuses on a case study with the aim of investigating the ability of cloud-
resolving grid spacing atmospheric simulations to capture the main features of an
event causing a very severe flash flood. These simulations are performed using the
Weather Research and Forecasting (WRF, Skamarock et al. 2005) numerical
meteorological model forced by an ensemble of reanalysis fields from the 20th Century
Reanalysis Project (Compo et al. 2006, Compo et al. 2011). The work is also
important to reveal how well fine-scale models can simulate an event for which
observations used to initialize the forcing model are extremely sparse (see section 4).
One prior work, Michaelis and Lackmann (2013), showed some promising results in
the use of WRF for another historical event, the New England Blizzard of 1888, but
that event was a midlatitude cyclone driven by dynamics on a larger-scale. More on
the windstorm modelling side, Stucki et al. (2015) reconstructed a 1925 high-impact
foehn storm in the Swiss Alps.
In this study, the case under investigation was a very intense flash-flood producing
event that occurred in 1915 in eastern Liguria (20-25 km east of Genoa, Liguria
region capital city), affecting San Fruttuoso, a small hamlet near Portofino, and the
coastal cities of Santa Margherita Ligure, Rapallo, and Chiavari (Figure 1). Based on
the newspapers of the time and documentary sources, after relatively light rain during
the night between September 24th and 25th, on the early morning of September 25th,
the area was hit for a few hours (7-11 UTC) by violent rain that triggered widespread
flash flooding, and a devastating debris flow. This landslide half-demolished the San
Fruttuoso thousand-year old abbey and laid down a thick layer of sand and rocks to
form a still existing 20-metre-wide 2-metre-deep beach (Faccini et al. 2008),
nowadays a very popular seaside resort. Based both on the observations of the time
(wind speed/direction, rainfall, observed lightnings) available for north-western Italy,
and on the model simulations, the occurrence of a back-building MCS is suggested.
*Fig. 1*





The paper is organized as follows. In Section 2 the 1915 convective event is
presented. Section 3 describes the WRF-ARW model setting performed. Results are
discussed in Section 4. Conclusions are drawn in Section 5.

**2.  Meteorological scenario**
The synoptic and mesoscale information for this event are available both from the 20[th]
Century Reanalysis Project (Compo et al. 2006, Compo et al. 2011) and from the
weather bulletins issued on a daily basis by the Italian Royal Central Office for
Meteorology (Regio Ufficio Centrale di Meteorologia e Geodinamica).
The 20[th] Century Reanalysis Project is an effort led by the Earth System Research
Laboratory (ESRL) Physical Sciences Division (PSD) of the National Oceanic and
Atmospheric Administration (NOAA) and the Cooperative Institute for Research in
Environmental Sciences (CIRES) at the University of Colorado to produce a reanalysis
dataset covering the entire twentieth century, assimilating only surface observations
of synoptic pressure, monthly sea surface temperature and sea ice distribution. The
observations have been assembled through international cooperation under the
auspices of the Atmospheric Circulation Reconstructions over the Earth (ACRE)
initiative, and working groups of Global Climate Observing System (GCOS) and World
Climate Research Program (WCRP). The Project uses an Ensemble Filter data
assimilation method, which directly yields each six-hourly analysis as the most likely
state of the global atmosphere, and gives also estimates of the uncertainty in that
analysis. This dataset provides the first estimates of global tropospheric variability
spanning from 1851 to 2012 with a six-hourly temporal resolution and a 2.0° grid
spacing. This study adopts 20th Century Reanalysis Project version 2C, which uses the
same model as version 2 with new sea ice boundary conditions from the COBE-SST2
(Hirahara et al. 2014), new pentad Simple Ocean Data Assimilation with sparse input
(SODAsi.2) sea surface temperature fields (Giese et al. 2016), and additional
observations from ISPD version 3.2.9 (Cram et al. 2015, Compo et al. 2013, Hirahara
et al. 2014, Krueger et al. 2013, Whitaker et al. 2004).
The weather bulletins issued by the Italian Royal Central Office for Meteorology
include weather maps at 7 UTC and 20 UTC and data (sea level pressure, wind
(direction and speed), temperature, cloud cover, cloud direction, state of the sea,
weather of the past 24 hours and notes) from about 125 Italian stations.
According to the reanalysis fields, the baroclinic circulation over Europe at 6 UTC of
September 25[th], (i.e. a few hours before the most intense phase of the event) is quite
typical for heavy precipitation events over the study area, with an upper-level trough
over Great Britain leading to a diffluent flow over the Liguria sea area, in combination
with a widespread high pressure block on eastern Europe and southern Russia (Fig.
2a). The diffluent flow over the Liguria sea area is associated with warm air advection
at 850 hPa from the southern Mediterranean towards northern-western Mediterranean
coastlines (Fig. 2c). Further information is provided by the mean sea level pressure
(MSLP) field at the European scale: both the reanalysis field (06 UTC, Fig. 2b) and the
Italian weather map (7 UTC, Fig. 3) show an elongated trough over the western
Mediterranean and a prominent ridge over south-eastern Europe, representing a
blocking condition on the large-scale. The Italian weather map gives also evidence of
a significant surface pressure gradient between the Po Valley and the Liguria sea.

*Fig. 2*

*Fig. 3*

On the mesoscale, at 06 UTC, a significant 2-metre temperature difference, around 3-
4 °C, is apparent from 20[th] Century Reanalysis Project fields between the Po Valley





and the Liguria sea (Fig. 4a), as well as a significant 2-metre specific humidity
gradient (Fig. 4b). The temperature difference is also confirmed by the available
observations at 07 UTC provided the Italian Royal Central Office for Meteorology (Fig.
4c).
*Fig. 4*

These mesoscale features represent the necessary ingredients for the generation of a
back-building MCS offshore of the Liguria coastline, as observed in the 2010, 2011
and 2014 high impact weather events in this region (Parodi et al. 2012, Rebora et al.
2013, Fiori et al. 2014).
The back-building MCS hypothesis is supported by the 48-hour quantitative
precipitation estimates (QPEs) for the period 24th September 07UTC - 26th September
07UTC (Fig. 5). The raingauges (64) contributing to this map have been provided by
different datasets such as the European Climate Assessment & Dataset project (Klein
Tank et al. 2002, Klok and Klein Tank 2009), the KNMI Climate Explorer dataset
(Trouet and Van Oldenborgh 2013), the Italian Meteorological Society (SMI, Auer et
al. 2005), the Piedmont Region climatological dataset (Cortemiglia 1999), and the
Chiavari Meteorological Observatory (Ansaloni 2006).
*Fig. 5*


The QPE map shows clearly a v-shaped elongated pattern, very similar to the ones
observed for the aforementioned events in Liguria. Based on historical information on
sub-daily rain rates, it can be estimated that during the most intense phase of the
event, the  rainfall depths reached up to 400 mm in approximately 4 hours (7-11 UTC
on September 25th) in some raingauges (Faccini et al. 2009): as a consequence of this
intense and highly localized rainfall the coastal cities of Rapallo, Santa Margherita
Ligure, Chiavari and San Fruttuoso suffered very serious damages (Fig. 6), with a
death toll around 25-30 people. Interestingly, as in the case of the Genoa 2014 event
(Lagasio et al. 2016) a very intense lightning activity was documented by the Italian
Royal Central Office for Meteorology (Fig. 7).
*Fig. 6*

*Fig. 7*


**3. WRF-ARW model simulations**
The model simulations have been performed using the Advanced Research Weather
Research and Forecasting Model (hereafter as ARW-WRF, version 3.4.1). Initial and
boundary conditions were provided by the 20th Century Reanalysis Project Version
version 2c (Compo et al. 2006, Compo et al. 2011) The ARW-WRF model was applied
for each of the 56 members of the ensemble provided by the 20th Century Reanalysis
Project database.
The ARW-WRF model is configured for this case study based on the results achieved in
the WRF modelling of the Genoa 2011 and Genoa 2014 v-shape convective structures
(Fiori et al. 2011, Fiori et al., 2015). Three nested domains (Fig. 8), centered on the
Liguria region, were used with the outer nest d01 using 25 km horizontal grid spacing



(61x55 grid points), the middle nest d02 using 5 km grid spacing (181x201 grid
points) and the innermost nest d03 using 1 km grid spacing (526x526 grid points).
The benefits of a high number of vertical levels have been demonstrated in Fiori et al.
(2014), and thus the same higher number of vertical levels (84) is adopted in this
study. Since the grid-spacing ranges from the regional modelling limit (25 km) down
to the cloud resolving one (1 km), two different strategies have been adopted with
regard to convection parameterization. For the domain d01 we adopted the new
simplified Arakawa–Schubert scheme (Han and Pan 2011) as it is also used by the
20[th] Century Reanalysis Project with 2.0° grid spacing. Conversely, a completely
explicit treatment of convective processes has been carried out on the d02-5 km and
d03-1 km domains (Fiori et al., 2014).

***Fig. 8***


The double-Moment Thompson et al. (2008) scheme for microphysical processes has
been adopted: this scheme takes into account ice species processes, whose relevance
in this case study is confirmed by the intense lightning activity observed during the
event, by modelling explicitly the spatio-temporal evolution of the intercept parameter
$N_i$ for cloud ice. Furthermore, the Thompson scheme was shown to be the best
performing for the Genoa 2011 and Genoa 2014 studies (Fiori et al. 2014 and 2015).
With regard to the results in Fiori et al. (2014) about the role of the prescribed
number of initial cloud droplets -$Nt_c$- created upon autoconversion of water vapour to
cloud water and directly connected to peak rainfall amounts, a maritime value
corresponding to a $Nt_c$ of $25*10^6\,m^{-3}$ has been adopted.
It is important to highlight that the availability of the 56 members ensemble is a key
strength in the present study, which enables estimates of uncertainties associated
with dynamical downscaling down to the WRF d03-1 km domain.

**4. Results and discussion**

A fundamental ingredient for the occurrence of back-building MCSs is the presence of
a persistent and robust convergence line: the availability of a large 1 km WRF
dynamically downscaled ensemble (56 members) allows the exploration of how many
members produce such a convergence line over the northern part of the Liguria sea
region where most of such MCSs form (Rebora et al. 2013). A convergence line is
here classified as persistent and robust if the minimum value of the divergence within
the study area is less than $-7*10^{-3}\,s^{-1}$ for at least 4 hours in a row. The divergence
threshold equal to $-7*10^{-3}\,s^{-1}$ corresponds to the 99.95% percentile of the divergence
values computed in every grid point within the region 7.50-10.25E / 43.75-44.50N in
Fig. 8 for each ensemble member in the period 12UTC 24[th] September – 00UTC 26[th]
September  (with a 30-minute time resolution).

Using the above threshold, 17 of the 56 WRF-ARW runs exhibit a persistent and
robust convergence line in the considered period. In particular, the time series of
divergence for four members (1, 13, 22, and 37 respectively) show that the minimum
is reached (Fig. 9 at approximately at the same time hourly QPF exceeds 50 mm/h
(Fig. 10, panels a-d, and g-l, members 1 and 13, Fig. 13, panels a-d, and g-l,
members 22 and 37; the other 13 members are not shown as they behave very
similarly). The four representative members exhibit also large QPFs over the whole 36
hours of the simulations (Fig. 10, panels f and n, members 1 and 13, Fig. 11, panels f



and n, members 22 and 37), even though significant differences both in the total
amount and in the spatial distribution are found. Significant values of the Lightning
Potential Index (LPI, Yair et al. 2010), in good agreement with the observations of the
Italian Royal Central Office for Meteorology are shown in Fig. 10 (panels e and m,
members 1 and 13) and Fig. 11, (panels e and m, members 22 and 37).
*Fig. 9*
*Fig. 10*
*Fig. 11*
Yet, most of the back-building MCS-producing members are affected by a non-
negligible location error (see panel 6 of Figures 10 and 11 for the four selected
members) with respect to the observed daily rainfall map (Fig. 5). This feature is
largely due to a predominance of the south-easterly wind component over the north-
westerly one (coming from Po Valley), thus pushing the convergence line too north-
westwards, close to the western Liguria coastline. This discrepancy is explained by the
highly localized spatio-temporal nature of this event, by the comparatively low spatial
density of the surface pressure stations assimilated by the 20$^{th}$ Century Reanalysis
Project over the western Mediterranean region (Fig. 12) and by the relatively coarse
characteristics (2.0° grid spacing, and 6-hourly temporal resolution) of the 20$^{th}$
Century Reanalysis Project forcing initial and boundary conditions data. For instance,
the primary wind convergence area over the sea and the inland area affected by the
rainfall (6.5-10.5° E / 43.5-45.5° N) is represented by only a few (2-3) 20$^{th}$ Century
Reanalysis Project grid points.
*Fig. 12*
To quantitatively examine precipitation errors for each WRF-ARW ensemble member,
a bias and mean absolute error (MAE) analysis of the 36 hour (12UTC 24/09 – 00UTC
26/09) QPF versus the 48 hour QPE (07UTC 24/09 – 07UTC 26/09) is undertaken by
comparing the available 64 raingauges with the nearest grid points of the d03-1 km.
The use of different time periods for QPE and QPF is not an issue as most of the
observed precipitation reported for Liguria fell in a time span encompassed in the run
time of the simulations. The results (Fig. 13) show that most of the 56 WRF members
have a negative BIAS of roughly 10-40 mm, largely explained by the ensemble
widespread underestimation of the extreme rainfall depths over the coastal cities of
Santa Margherita Ligure, Rapallo, and Chiavari. The 17 selected members (red
markers) show an average BIAS of -22 mm and a MAE of 40 mm, while the remaining
39 members have an average BIAS of -31 mm and a MAE of 42 mm. Also for the 17
selected members, the BIAS is largely explained by the stations mostly affected by
the MCS and it reduces to -8 mm when Chiavari, Cervara and S. Margherita Ligure are
excluded from the comparison.
*Fig. 13*
Because traditional verification measures applied to QPF are greatly influenced by
location errors, a deeper understanding of QPF performance in the WRF ensemble is
gained by performing object based verification using the Method for Object-based
Diagnostic Evaluation (MODE, Davis et al. 2006a, 2006b), intended to reproduce a
human analyst's evaluation of the forecast performance. The MODE analysis is
performed using a multi-step automated process. A convolution filter is applied to the
raw field to identify the objects. When the objects are identified, some attributes



regarding geometrical features of the objects (such as location, size, aspect ratio and
complexity) and precipitation intensity (percentiles, etc.) are computed. These
attributes are used to merge objects within the same forecast/observation field, to
match forecast and observed objects and to summarize the performance of the
forecast by attribute comparison. Finally, the interest value combines in a total
interest function the attributes (the centroid distance, the boundary distance, the
convex hull distance, the orientation angle difference, the object area ratio, the
intersection divided by the union area ratio, the complexity ratio, and the intensity
ratio) computed in the object analysis, providing an indicator of the overall
performance of matching and merging between observed and simulated objects.  In
the present study, the relative weight of each attribute used the default setting in MET
(Halley Gotway et. al. 2013). The displacement errors including centroid distance and
boundary distance were weighted the greatest in the calculation of total interest.
In our experiment we have empirically chosen the convolution disk radius and
convolution threshold, so that this choice would recognize precipitation areas (at least
roughly 50x50 km or so) similar to what a human would identify. For each WRF
ensemble member the 36-hour (12UTC 24/09 – 00UTC 26/09) QPF is compared with
the 48-hour QPE (07UTC 24/09 – 07UTC 26/09), both bilinearly interpolated to the
same 10 km grid. This grid spacing represents a good compromise between the native
1 km WRF grid spacing and the 40 km average distance between the available 64
raingauges. After a set of experiments, we fixed the value of the convolution radius to
one grid point and the threshold of the convoluted field to 75 mm. Twelve members
out of the 17 members selected using the minimum divergence criterion show
significant values (above 0.8) of the total interest function. This value is slightly
higher than the default one (0.7) used by MODE to match paired objects, in order to
restrict our analysis to the best simulated events. Selected members 1, 13, 22 and 37
(Fig. 14) have total interest values above 0.93 (close to 1 is good) and their paired
clusters distance, namely the distance between centroids of observed and simulated
rain regions, is around 100 km. Furthermore, the area ratio -that provides an
objective measure of whether there is an over- or underprediction of the areal extent
of the forecast- ranges between 0.80 and 1.1, suggesting a reasonable agreement
with observations. However, the differences are larger for the median (50th percentile)
and near-peak (90th percentile) rainfall values: the predicted values are 30% lower
than the observed ones, suggesting an overall underestimation of the intense rainfall
observed.

*Fig. 14*

## 5. Conclusions

Highly localized and persistent back-building MCSs represent one of the most
dangerous flash-flood producing storms in the north-western Mediterranean area. A
historic extreme precipitation event occurring over Liguria on September 1915, which
seems to be due to one of these systems, was investigated in this paper both by
means of a large collection of observational data and by means of atmospheric
simulations performed using the WRF model forced by an ensemble of reanalysis
fields from the 20th Century Reanalysis Project.
The results show that the simulated circulation features are consistent with the
hypothesis of a highly localized back-building MCS over Liguria sea, and that the WRF
runs -driven by a significant fraction of the members of the 20th Century Reanalysis



Project ensemble- produce fields that are in reasonable agreement with the observed
data.
The proposed approach was to focus only on the WRF runs showing strong
convergence so as to get the best depiction of the event. Thus, we suggest that, when
using datasets such as the 20th Century Reanalysis Project, it is important to consider
that the physics/dynamics are likely to play a role in the events of interest, and to
follow a similar technique to selectively use the Reanalysis ensemble members best
displaying the key physics/dynamics of the event. Future work should test further an
approach like this one to get a better understanding of how well the same
convergence detection approach in regional climate model simulations of past and
future climate (e.g. Pieri et al. 2015 at cloud-permitting grid spacing) can quantify
possible changes in back-building MCS precipitation processes.
On the data collection side, this study showed that in addition to the use of Reanalysis
products, other sources of data, such as newspapers, photographs, and historical
meteorological bulletins can be essential sources of knowledge. Focusing on historical
meteorological bulletins, future work on this particular case and similar ones occurring
along the north-western Mediterranean coastline will explore the use of  bogus
observations or other preprocessing techniques to alter lower tropospheric conditions
at model initialization time to better match actual observations, which may result in a
better location of the convergence line and consequently simulation of the
precipitation event.

## 6. Acknowledgments

This work was supported by the Italian Civil Protection Department and by the
Regione Liguria. The ground based observations were provided by Italian Civil
Protection Department and the Ligurian Environmental Agency. The raingauge data
were courtesy of the European Climate Assessment & Dataset project, the KNMI
Climate Explorer dataset, the Italian Meteorological Society, Piedmont Region
climatological dataset, and the Chiavari Meteorological Observatory. Antonio Parodi
would like also to acknowledge the support of the FP7 DRIHM (Distributed Research
Infrastructure for Hydro-Meteorology, 2011-2015) project (contract number 283568).
Thanks are due to the CINECA, where the numerical simulations were performed on
the Galileo System, Project-ID: SCENE. W. Gallus appreciates the opportunity for a
research visit at the University of Milan.

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





**Figures and figure captions**

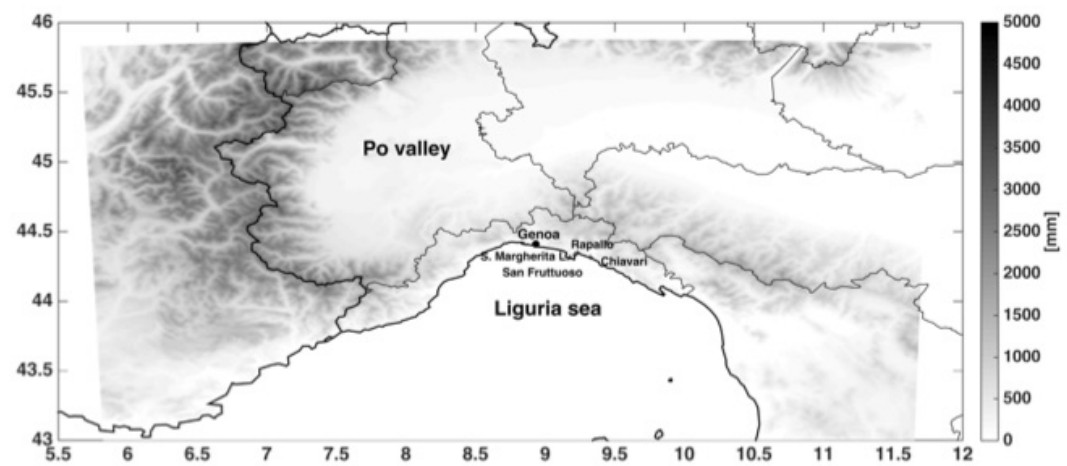

*Figure 1: Study region and Liguria coastal cities affected by the September*
*1915 event.*





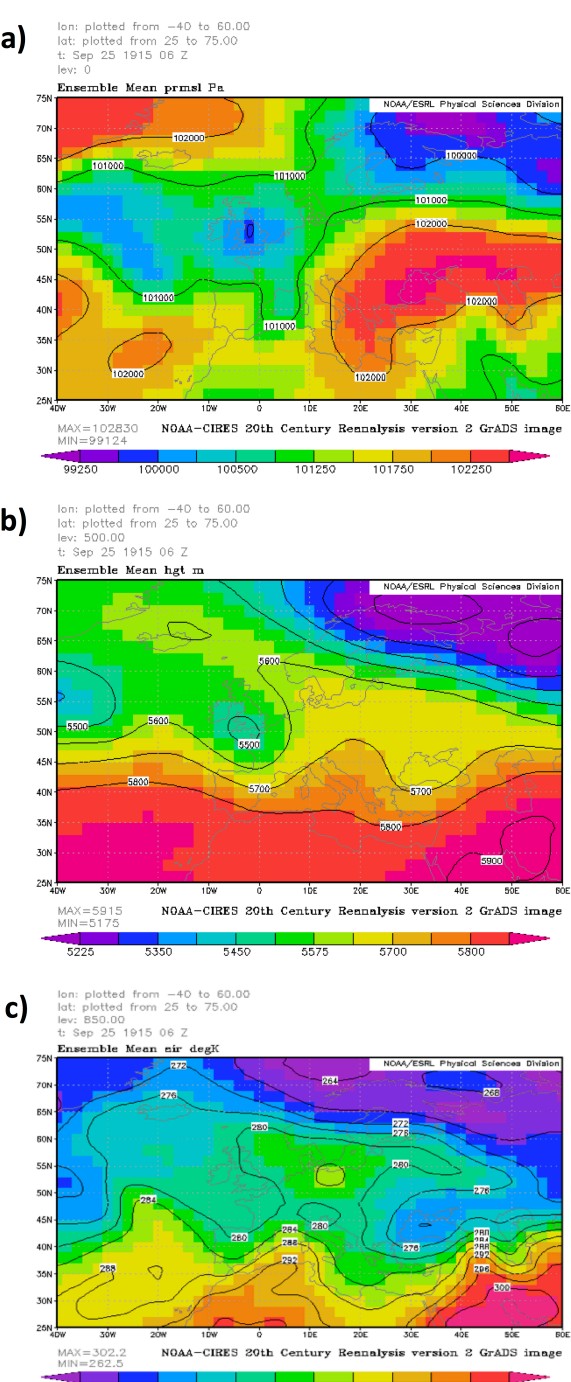

*Figure 2: a) sea level pressure, b) 500 hPa geopotential, and c) 850 hPa temperature on 25th September 1915 06UTC (20[th] Century Reanalysis Project mean fields over the 56 ensemble members).*




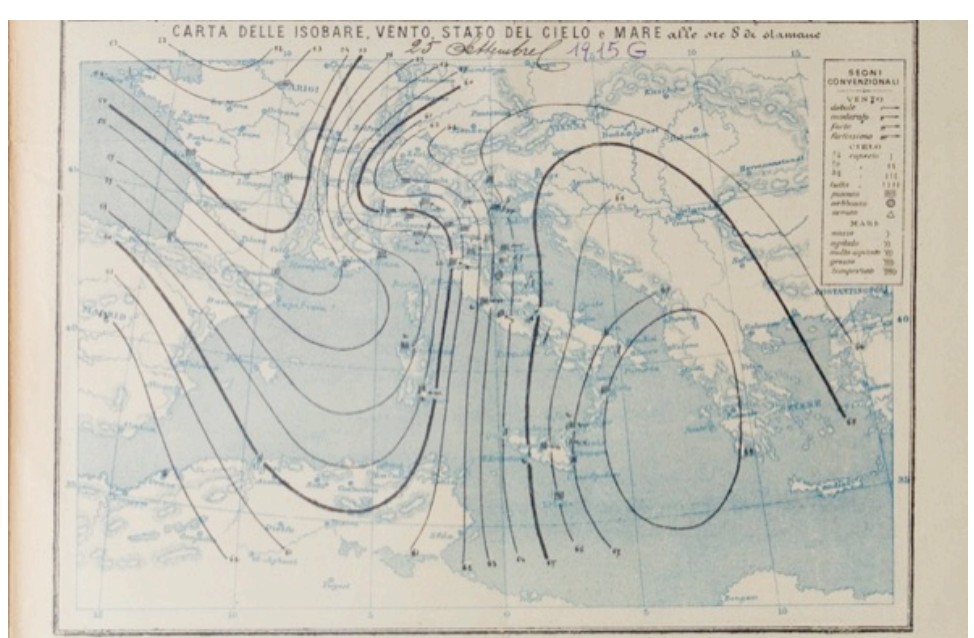


**Figure 3: surface pressure isobars on 25[th] September 1915 at 07UTC, as**
**provided by the Italian Royal Meteorological Service.**



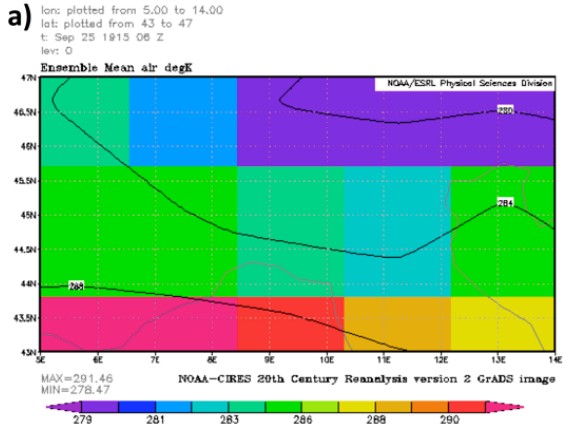

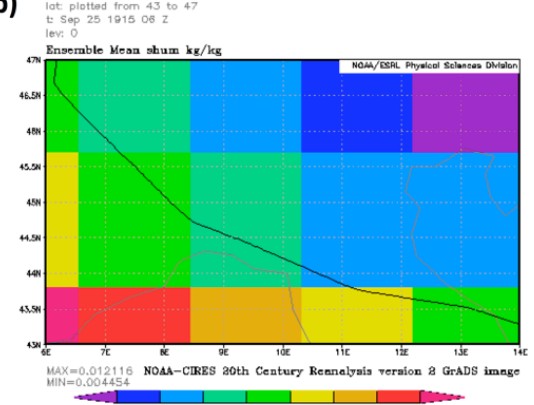


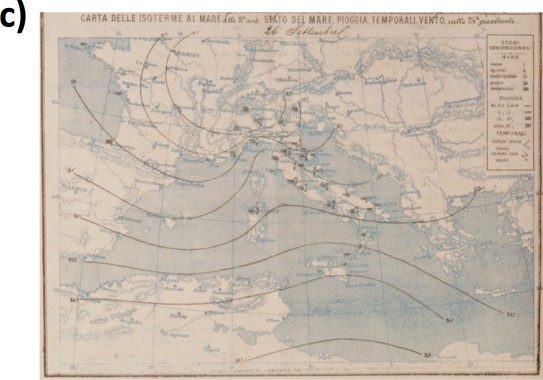


**Figure 4: a) 2 m temperature and b) 2 m specific humidity on 25th September 1915 06 UTC over the study region. (20th Century Reanalysis mean fields over the 56 ensemble members), c) surface temperature isotherms on 25th September 1915 at 07UTC, as provided by the Italian Royal Meteorological Service.**







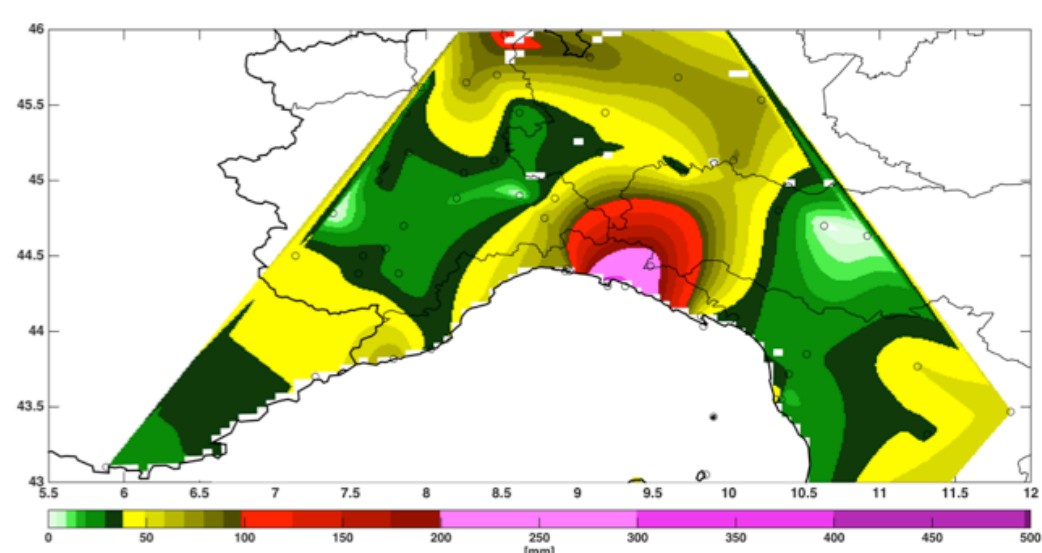

*Figure 5: quantitative precipitation estimates (QPE) for 24th September*
*07UTC - 26th September 1915 07UTC.*







*Figure 6: Rapallo flash-flood impacts on 25 september 1915 (Courtesy of real*
*estate Agency Bozzo in Camogli).*





***Figure 7: thunderstorms and lightning activity reports (red circle) on 25th***
***September 1915, as provided by the Italian Royal Meteorological Service.***






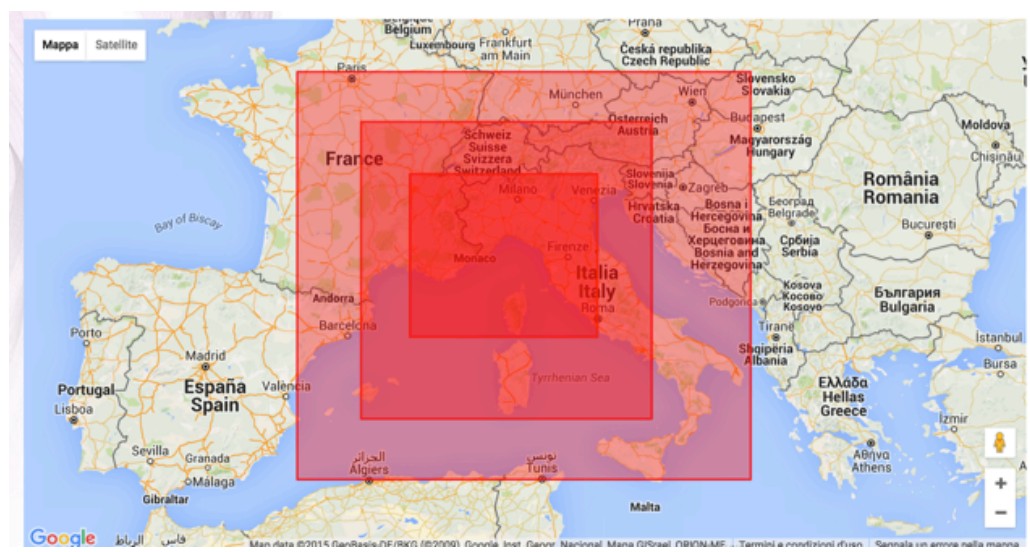

*Figure 8: domains for the numerical simulations of the Genoa 1915 event.*
*d01 (Δ=25 km), d02 (Δ=5 km) and d03 (Δ=1 km).*




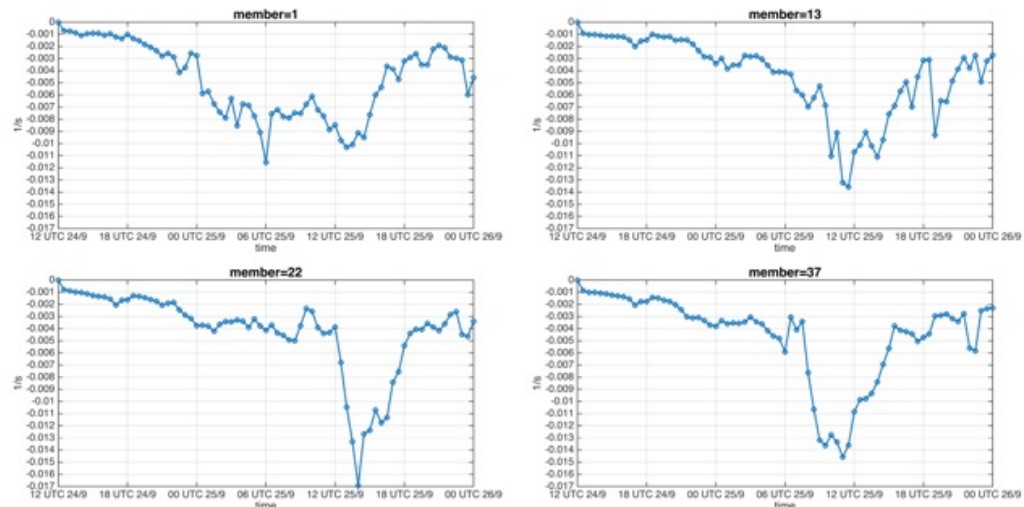



*Figure 9: minimum divergence time series for members 1, 13, 22 and 37.*



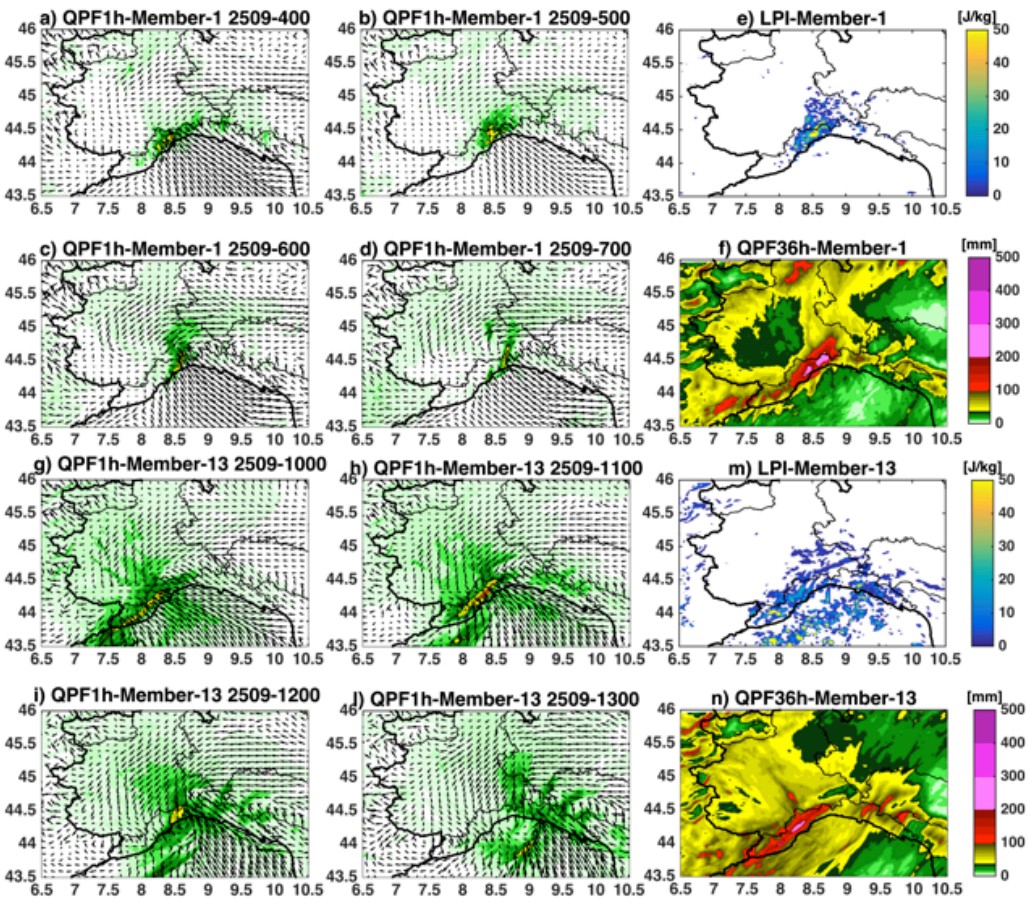

**Figure 10: Panels a-d, and g-l show the hourly QPF and 10 m wind fields corresponding to the period with the minimum divergence values in Figure 11 for members 1, and 13. Panels e-f, and m-n show the Lightning Potential Index accumulated over the same 4 hours period, and the 36 hour QPF, respectively for members 1, and 13.**



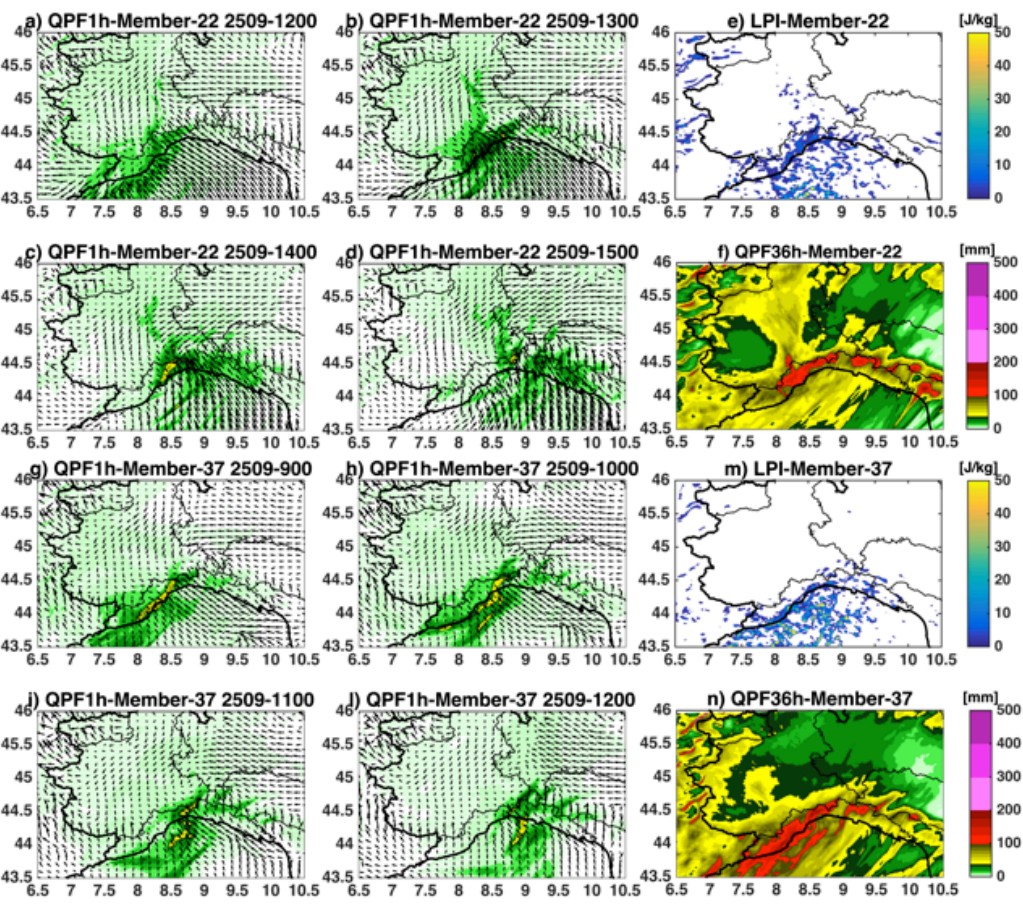

**Figure 11: Panels a-d, and g-l show the hourly QPF and 10 m wind fields corresponding to the period with the minimum divergence values in Figure 11 for members 22, and 37. Panels e-f, and m-n show the Lightning Potential Index accumulated over the same 4 hours period, and the 36 hour QPF, respectively for members 22, and 37.**





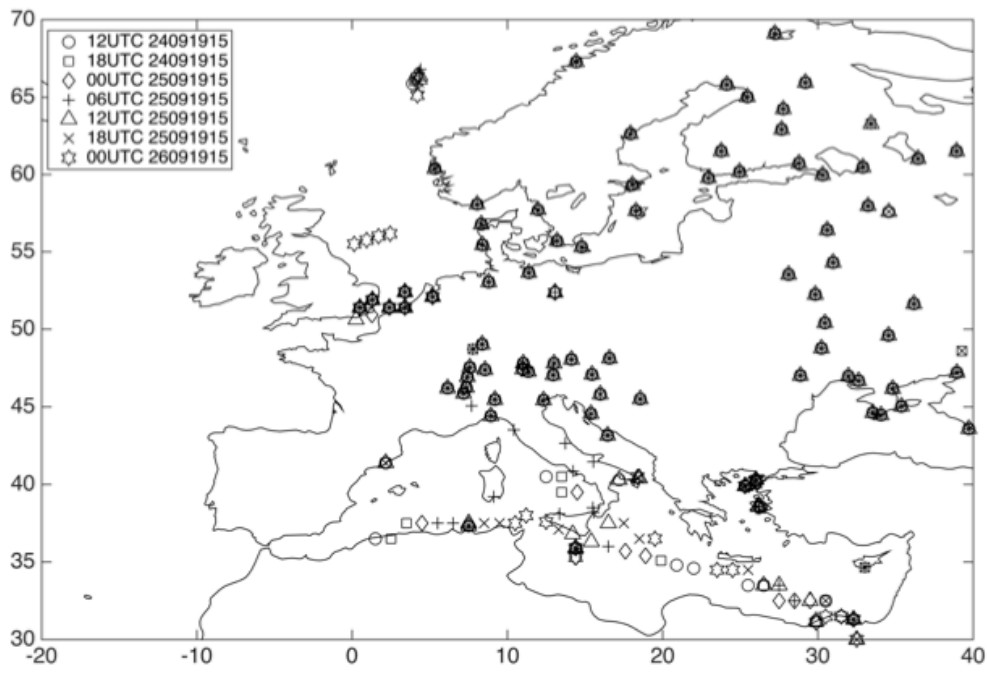

**Figure 12:  surface pressure stations assimilated every six hours in the**
**period 12UTC 24th September 1915 - 00UTC 26th September 1915.**






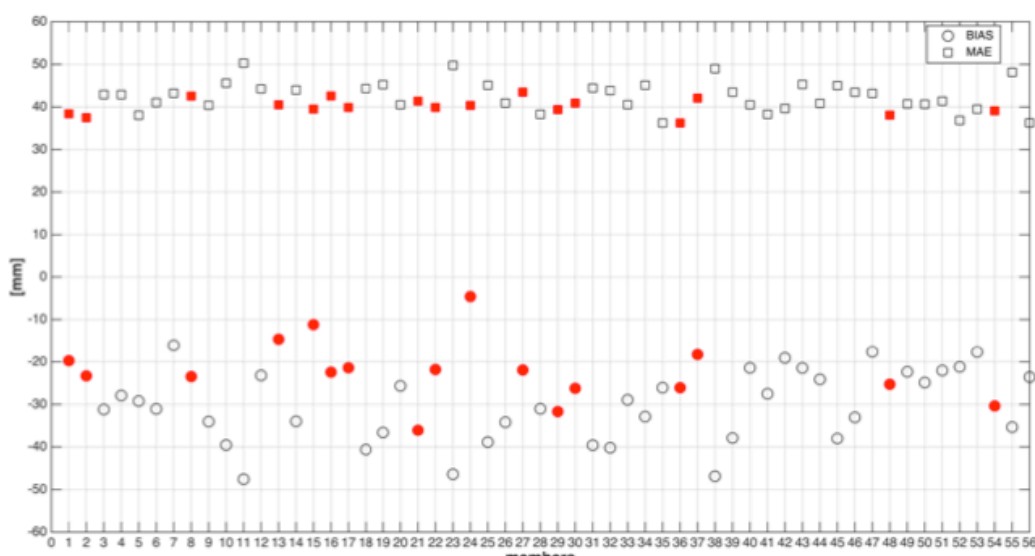

*Figure 13: rainfall depth BIAS and MAE for each d03-1km WRF member. Red*
*markers represent the 17 members producing robust and persisting*
*convergence lines over the Liguria Sea.*






**Figure 14:** **QPE regridded at 10 km grid spacing (panel a) and QPF from
members 1 (panel b), 13 (panel c), 22 (panel d) and 37 (panel e), regridded
at 10 km grid spacing (lower panels). Dots identify the areas of paired
clusters.**