# Peer review of "Ensemble cloud-resolving modelling of a historic back-building mesoscale convective system over Liguria: The San Fruttuoso case of 1915"

_Climate of the Past, 2016_

## Referee Comment (RC1) · Anonymous Referee #1 · 15 Dec 2016

The paper proposes to simulate, with the WRF model, the San Fruttuoso case of 1915 using the 20th Century Reanalysis ensemble as boundary conditions. Some model of the ensemble simulations are able to reproduce the strong convergence over the Liguria Sea and precipitation fields that are consistent with the occurrence of highly localized and persistent back-building mesoscale convective systems. The results are also supported by unconventional data such as historical meteorological bulletins newspapers and photographs.

Major Points: 1. The 20th Century ensemble reanalysis at 2.0o is used as boundary

conditions for simulations with a 25km (∼0.22o) outer domain. In the analysis, the poor alignment of the convergence zones is attributed to the low resolution of the reanalysis but this could be mitigated by introducing a WRF domain with 125km. This should be attempted at least for the four members analysed. 2. The advantage of high resolution simulations is their ability to provide 3D information of an event. No analysis of the upper air results, vertical profiles or 2D vertical cross-sections has been presented. The dynamics of the storm evolution should be added to the manuscript. 3. Although an ensemble of 56 members is produced, only 2 deterministic measures of individual ensemble members are presented but no analysis of the quality of such ensemble is provided. A shortcoming of deterministic measures of skill is that information about prediction uncertainties is not available, thus categorical measures like Brier skill score, continuous ranked probability score, ROC skill score are a useful tool to assess the quality of an ensemble forecast. In the following references examples such types of analysis can be found. Please add some categorical measures.

4. The deterministic measures are also evaluated by comparing observations and simulations with different time spans. In lines 209-218 reference is made to rainfall depths for a 4 hour period thus QPE should be computed for the same time period as the simulation and only then should the evaluation be performed. In case that is not possible, the simulation should cover the same time period of the observations.

Corti, S., A. Weisheimer, T. N. Palmer, F. J. Doblas- Reyes, and L. Magnusson (2012), Reliability of decadal predictions, Geophys. Res. Lett., 39, L21712, doi:10.1029/2012GL053354 Wilks, D. S. (1995), Statistical Methods in the Atmospheric Sciences, 467 pp., Academic, San Diego, Calif. Doblas-Reyes, F. J., and Coauthors, 2009: Addressing model uncertainty in seasonal and annual dynamical ensemble forecasts. Quart. J. Roy. Meteor. Soc., 135, 1538–1559, doi:10.1002/qj.464.

Minor Points: • Figures 3 is very difficult to read. Since its quality cannot be enhanced I would suggest adding a figure 3b with the ensemble mean slp with the same domain and isobar resolution in order to better assess the resemblance between the

20th Century Reanalysis and the forecasted conditions on the 25th of September. The approximate pressure gradients in the Po Valley, Mediterranean and France, in both analysis, would be appreciated. • Figure 4c is as difficult to read as figure 3. Figures 4a and b should represent the same domain as figure 4c. Same argument as before. • In lines 209-218 reference is made to rainfall depths for a 4 hour period. If sub-daily precipitation is available please add either QPE or individual stations time series for the periods analysed in figures 10 and 11. • The topography of the WRF plays a fundamental role in the development of the convective system but is missing from the manuscript. I suggest replacing the map in figure 8 with the model topography for all the domains. To facilitate the comparison with the real topography, I would suggest the merger with figure 1 as figure 1b. Also in figure 1 there is no reference to the source of the topographic map. Lines 95-100 – Paragraph is too long, please rephrase. Line 170 – The paragraph refers to 500hPa chart, i.e. figure 2b Line 178 – Should be figure 2a, not 2b Line 746 – Y axis in figure 9 difficult to read. Reduce the resolution and increase the caption font. Line 755 – Indexation of figure 10 and 11, hard to follow. Attribute the indices sequentially. Legend should describe better the individual panel figures.

––––––––––––––––––––––––

---

## Referee Comment (RC2) · Anonymous Referee #2 · 24 Dec 2016

This work shows the benefits of using numerical methods combined to historical documents to offer a detailed insight into an event that it's not only historically relevant but it's also still observed in the present days. It is commendable that the authors take into account the 56 members ensemble of the 20th century reanalysis project version 2c for members dynamics exploration given the sparse surface data in 1915. Results are shown for the four members reproducing the best strong convergence over the Liguria sea.

General comments:

[Figure]

1. Even if 56 members on the 20th century reanalysis were studied, only four of them reproducing the best the event's dynamics were taken into account while showing the results. It would be interesting to have some comments about the members showing very "non-realistic dynamics" and also about the mean ensemble.

2. Convective systems are generally associated with vertical motion. WRF outputs offers 3D information allowing the generation of vertical cross-section plots or Skew-T diagrams, none of them are shown in the paper. Some graphs and words about this should be added.

3. In general the writing style and content is of good quality but the graphs are not at the same level of quality. Fig. 2 has a background hard to see, Fig. 3 has low quality, Fig. 8 is upgradable, etc. (Check the Specific comments).

4. While the convergence line is a very important criteria for dynamics exploration, it hasn't been shown in any figure. Lines 273 and 274 signals the coordinates of this line but a graphical representation would clarify it.

Specific comments:

- L113 cites WRF version 2 while the work uses WRF version 3, the correct citation would be thus Skamarock et al. 2008 (NCAR/TN–475+STR)

- L128 shows a good example in dates using sometimes upper-case and not using this. This is reproduced all over the paper. Consistency in the style should be shown.

- L179 makes reference to Fig. 2b where it's shown 500hPa Geopotential but this is not stated in the text. Please add a comment on this field.

- L280 text makes reference to QPF even if this abbreviation hasn't been introduced. Please define it.

- L281 addresses Fig. 13 while it should be Fig. 11

- L296 mentions a panel 6 which it's not shown in Fig. 10

- L316 justifies the use of different periods for QPE and QPF but it should be possible to take into account the same period from simulations.

- L347 uses the acronym MET even if it hasn't been defined. Please define it.

- Fig. 2 has a background difficult to read an thus it's hard to identify the location of the fields. Also Fig 2b is not being used in the text correctly.

- Fig. 3 could be maybe digitized for better quality

- Fig. 4 uses different domains and colours are hard to interpret.

- Fig. 8 should show better the topography as seen by the WRF model and also the convergence line grid points over the Liguria sea. It also calls it the "Genoa 1915 event" even if the paper states "The San Fruttuoso Case of 1915"

---

## Author Comment (AC1) · 4 Feb 2017

We thank referee 1 for his/her positive comments on the topic of the manuscript and the analysis we carried out and for the many useful suggestions that will help us in preparing an improved version of the manuscript. In the following, we address his/her comments.

Main comments

1. The 20th Century ensemble reanalysis at 2.0 degrees is used as boundary conditions for simulations with a 25km (âĹij0.22 degree) outer domain. In the analysis,

the poor alignment of the convergence zones is attributed to the low resolution of the reanalysis but this could be mitigated by introducing a WRF domain with 125km. This should be attempted at least for the four members analysed.

Reply: It is true that the ratio of the grid spacing for the driving data to that of our outer 25 km domain (roughly 1:9) is larger than what is normally recommended, and larger than the 1:5 ratio we use for our inner domains. Because of the large computational cost to run 56 ensemble members with each one including a refined 1 km domain, the decision originally had been made to only use 3 nests. Because we no longer have access to such a large amount of computer resources, we are unable to re-run all 56 members, but we recognize that the reviewer raises an interesting question. Therefore, with the limited resources remaining, we have chosen to re-run the 4 best members (1, 13, 22 and 37) adding a fourth outer domain with 125 km grid spacing (in addition to 25, 5, and 1 km grid spacing) (see Fig. 1). The comparison of 36 hour QPF for the innermost domain at 1 km grid spacing is provided in Figure 2: first row results corresponding to simulations driven by outermost domain at 25 km grid spacing, second row the same but with outermost domain at 125 km grid spacing. Heavier areal QPF can be seen in all members in the first row (our original configuration), both on the entire 1 km grid spacing domain (both on sea and land areas), and also on the smaller area over which the paper focuses. This statement is confirmed when comparing, for the 4 selected best members, the BIAS and MAE (in mm) over the available 64 raingauge stations in the runs using an outermost domain with 25 and 125 km grid spacing respectively.

The New Simplified Arakawa-Schubert (NSAS) scheme adopted in these additional simulations over the 125 and 25 km grid spacing domains has been revised, for deep moist convection, to make cumulus convection stronger and deeper to deplete more instability from the atmospheric column and result in the suppression of excessive grid-scale precipitation (Han and Pan, 2011). This can result, if applied even at very coarse grid spacing (125 km), in an overall reduction of the efficiency of the precipitation pro-

Interactive
comment

**Table 1.** BIAS and MAE (in mm) for the 4 selected best member ensembles over the available 64 raingauge stations in the runs having an outermost domain with 25 and 125 km grid spacing, respectively.

| member | BIAS-d01 25 km | BIAS-d01 125 km | MAE-d01 25 km | MAE-d01 125 km |
|---|---|---|---|---|
| 1 | -19.8 | -30 | 38.4 | 38.9 |
| 13 | -14.6 | -26.4 | 40.5 | 42.2 |
| 22 | -21.8 | -29.8 | 39.9 | 45.9 |
| 37 | -18.2 | -26.9 | 42.1 | 44.6 |

cesses, thus impacting also the results on the innermost domains down to 1 km grid spacing. We therefore believe it is advantageous to maintain our previously obtained results and to not introduce the extra 125 km domain, which adversely affects results, likely because of the NSAS scheme that we use.

2. The advantage of high resolution simulations is their ability to provide 3D information of an event. No analysis of the upper air results, vertical profiles or 2D vertical cross-sections has been presented. The dynamics of the storm evolution should be added to the manuscript.

Reply: the physical mechanism responsible for the generation of the back-building MCS observed on 25 september 1915 also has been recently explained by Fiori et al. (2016). Taking advantage of the availability of both observational data and modelling results at the micro-$\alpha$ meteorological scale, Fiori et al. (2016) provide insights about the triggering mechanism and the subsequent spatio-temporal evolution of the Genoa 2014 back-building MCS. The major finding is the important effect of a virtual mountain created on the Ligurian sea by the convergence of a cold and dry jet outflowing from the Po valley and a warm and moist low level south-easterly jet within the PBL. The same mechanism is active also for this case. Let us consider, as an example, the convective flow field at 06UTC on 25 september 1915 (see Figure 3), as predicted by member 1 of the ensemble. Panel A shows the 2 m potential temperature field together with the

10 m horizontal wind vector field: the colder and drier jet outflowing from the Po valley and the warmer and moister air from southern mediterranean sea are evident. Panel B shows, by means of the potential temperature along the cross section corresponding to the green dotted line of Panel A, also the thin potential temperature layer (virtual mountain) in front of the actual Liguria topography. This acts, in agreement with Fiori et al. (2016), to produce the strong convective cells in panel C (updraft velocity above 10 m/s) with the apparent back-building on the western side (less mature and intense cells around 8.4° latitude). The main updraft produces vertical advection of water vapor (panel D), thus resulting in significant production of rainwater (panel E), snow (panel F, significantly advected inland by the upper level south-westerly winds), and graupel (panel G). We will present this analysis in the revised version of the manuscript.

3. Although an ensemble of 56 members is produced, only 2 deterministic measures of individual ensemble members are presented but no analysis of the quality of such ensemble is provided. A shortcoming of deterministic measures of skill is that information about prediction uncertainties is not available, thus categorical measures like Brier skill score, continuous ranked probability score, ROC skill score are a useful tool to assess the quality of an ensemble forecast. In the following references examples such types of analysis can be found. Please add some categorical measures.

Reply: It is well know (Mass et al. 2002) that point-to- point verification measures like those usually used for traditional ensemble verification do not work well with fine grid spacing simulations, because a double penalty exists for spatial errors, which are extremely common for high intensity precipitation events. This problem is likely even worse when limited observations from 1915 are used. Object-based verification techniques have been developed in the last 10-15 years specifically because of these problems. The application of the MODE Object-based verification technique showed that twelve members out of the 17 members selected using the minimum divergence criterion have significant values (above 0.8) of the total interest function. More interestingly these twelve members perform well in some of the MODE-derived statistics presented

**Table 2.** CENTROID DISTANCE: provides a quantitative sense of spatial displacement of forecast. FCST AREA/OBS AREA: provides an objective measure of whether there is an over-or under-prediction of areal extent of forecast. FCST INT 50/OBS INT 50 and FCST INT 90/OBS INT 90 provide objective measures of Median (50th percentile) and near-Peak (90th percentile) intensities found in objects. TOTAL INTEREST: provides summary statistic derived from fuzzy logic engine with user-defined interest maps for all these attributes plus some others.

| Parameter | Average | Standard deviation |
|---|---|---|
| PAIRED CENTROID DISTANCE (km) | 114 | 62 |
| FCST AREA/OBS AREA | 1.10 | 0.90 |
| FCST INT 50/OBS INT 50 | 0.73 | 0.06 |
| FCST INT 90/OBS INT 90 | 0.62 | 0.11 |
| TOTAL INTEREST | 0.88 | 0.09 |

in the table below: in particular it is impressive that we obtain very satisfactory displacement errors with the very crude initialization of a 1915 reanalysis case.

For instance, the paper of Duda and Gallus (2013) found an average displacement distance (absolute error) for initiation of systems to be 105 km. Squitieri and Gallus (2016) show in Figs 7 and 8 the centroid of all MCSs forecasted by WRF and the centroid of observed MCSs for a sample of 15 strongly forced and 16 weaker forced events, and the distance between the averaged centroids is usually 100 km or more. Similar consideration also holds for the ratio between the near-peak (90th percentile) intensities found in the observed and predicted objects. We will present this information in the revised version of the manuscript.

4. The deterministic measures are also evaluated by comparing observations and simulations with different time spans. In lines 209-218 reference is made to rainfall depths for a 4 hour period thus QPE should be computed for the same time period as the simulation and only then should the evaluation be performed. In case that is not possible, the simulation should cover the same time period of the observations. Reply:

A comparison at an hourly level is basically meaningless (due to high variability within the simulations) and also impossible (no observational data are available on hourly scales). Furthermore the 12 hours covered by the observations and not covered by the model do not experience important precipitation (as supported with the notes on the past weather in the daily bulletins (e.g. "pioggia dal mattino" or "pioggia fino al pomeriggio"). Additionally the QPF in all 56 members for the period 12 UTC 24th – 00 UTC 25th is negligible over the entire Liguria Region (averaging below 1 mm in 12 hours). Therefore, verification statistics using the time periods we have chosen would not differ meaningfully from those performed if we had access to observations whose timing did match exactly the simulation period. We will mention this in the revised version of the manuscript.

Minor comments Figures 3 is very difficult to read. Since its quality cannot be enhanced I would suggest adding a figure 3b with the ensemble mean slp with the same domain and isobar resolution in order to better assess the resemblance between the 20th Century Reanalysis and the forecasted conditions on the 25th of September. The approximate pressure gradients in the Po Valley, Mediterranean and France, in both analysis, would be appreciated. Figure 4c is as difficult to read as figure 3. Figures 4a and b should represent the same domain as figure 4c. Same argument as before. In lines 209-218 reference is made to rainfall depths for a 4 hour period. If sub-daily precipitation is available please add either QPE or individual stations time series for the periods analysed in figures 10 and 11. The topography of the WRF plays a fundamental role in the development of the convective system but is missing from the manuscript. I suggest replacing the map in figure 8 with the model topography for all the domains. To facilitate the comparison with the real topography, I would suggest the merger with figure 1 as figure 1b. Also in figure 1 there is no reference to the source of the topographic map. Lines 95-100 – Paragraph is too long, please rephrase. Line 170 – The paragraph refers to 500hPa chart, i.e. figure 2b Line 178 – Should be figure 2a, not 2b Line 746 Y axis in figure 9 difficult to read. Reduce the resolution and increase the caption font. Line 755 – Indexation of figure 10 and 11, hard to follow. Attribute the

indices sequentially. Legend should describe better the individual panel figures.

Reply: we agree on the comments on the figures: in the revised version of the manuscript, we will reformat and reorganising them according to the suggestions from the reviewer. We will also correct the errors he/she highlighted and rephrase the paragraph at lines 95-100. Moreover, we will improve and clean up the captions of figures 10 and 11. It is however not possible to attribute the indices sequentially because the 6 panels in figure 10 and the 6 ones in figure 11 refer to different ensemble members. The only comment that we cannot fulfil is that concerning sub-daily precipitation as the hourly time series are not available. The reference to rainfall depth for a 4-hour period (the period corresponding to the maximum recorded precipitation) is available for just one station, but there are no hourly data.

References Duda, J. D., Gallus Jr, W. A. (2013). The Impact of Large-Scale Forcing on Skill of Simulated Convective Initiation and Upscale Evolution with Convection-Allowing Grid Spacings in the WRF. Weather and Forecasting, 28(4), 994-1018. Fiori, E., L. Ferraris, L. Molini, F. Siccardi, D. Kranzlmueller, and A. Parodi (2016), Triggering and evolution of a deep convective system in the Mediterranean Sea: modelling and observations at a very fine scale.. Q.J.R. Meteorol. Soc. Accepted Author Manuscript. doi:10.1002/qj.2977. Han, J., Pan, H. L. (2011). Revision of convection and vertical diffusion schemes in the NCEP global forecast system. Weather and Forecasting, 26(4), 520-533. Mass, C. F., Ovens, D., Westrick, K., Colle, B. A. (2002). Does increasing horizontal resolution produce more skillful forecasts?. Bulletin of the American Meteorological Society, 83(3), 407-430. Squitieri, B. J., Gallus Jr, W. A. (2016). WRF Forecasts of Great Plains Nocturnal Low-Level Jet-Driven MCSs. Part II: Differences between Strongly and Weakly Forced Low-Level Jet Environments. Weather and Forecasting, 31(5), 1491-1510
* * *
[Figure]

**Fig. 1.** ARF-WRF 4-domains setup.

---

## Author Comment (AC2) · 4 Feb 2017

We thank referee #2 for his/her positive comments on the topic of the manuscript and the analysis we carried out and for the many useful suggestions that will help us in preparing an improved version of the manuscript. In the following, we address his/her comments.

Main comments

1. Even if 56 members on the 20th century reanalysis were studied, only four of them reproducing the best the event's dynamics were taken into account while showing the

results. It would be interesting to have some comments about the members showing very "non-realistic dynamics" and also about the mean ensemble. Reply: we will consider this comment in revising the manuscript. Some information on the other members is indeed interesting, but we have also to keep the focus of the paper which investigates the ability of the ARF-WRF simulations to capture the MCS character of the event. The members showing very "non-realistic dynamics" and also the mean ensemble fail to capture the convergence line creation and its evolution responsible for the generation of the back-building MCS in 17 out of 56 members. We have therefore to balance the need of giving some information on the other members with the goal of keeping the focus of the paper on the back-building MCS character of the investigated event.

2. Convective systems are generally associated with vertical motion. WRF outputs offers 3D information allowing the generation of vertical cross-section plots or Skew-T diagrams, none of them are shown in the paper. Some graphs and words about this should be added. Reply: the physical mechanism responsible for the generation of the back-building MCS observed on 25 september 1915 also has been recently explained by Fiori et al. (2016). Taking advantage of the availability of both observational data and modelling results at the micro-$\alpha$ meteorological scale, Fiori et al. (2016) provide insights about the triggering mechanism and the subsequent spatio-temporal evolution of the Genoa 2014 back-building MCS. The major finding is the important effect of a virtual mountain created on the Ligurian sea by the convergence of a cold and dry jet outflowing from the Po valley and a warm and moist low level south-easterly jet within the PBL. The same mechanism is active also for this case. Let us consider, as an example, the convective flow field at 06UTC on 25 september 1915 (see Figure 1), as predicted by member 1 of the ensemble. Panel A shows the 2 m potential temperature field together with the 10 m horizontal wind vector field: the colder and drier jet outflowing from the Po valley and the warmer and moister air from southern mediterranean sea are evident. Panel B shows, by means of the potential temperature along the cross section corresponding to the green dotted line of Panel A, also the thin potential temperature layer (virtual mountain) in front of the actual Liguria topography. This acts,

in agreement with Fiori et al. (2016), to produce the strong convective cells in panel C (updraft velocity above 10 m/s) with the apparent back-building on the western side (less mature and intense cells around 8.4° latitude). The main updraft produces vertical advection of water vapor (panel D), thus resulting in significant production of rainwater (panel E), snow (panel F, significantly advected inland by the upper level south-westerly winds), and graupel (panel G). We will present this analysis in the revised version of the manuscript.

3. In general the writing style and content is of good quality but the graphs are not at the same level of quality. Fig. 2 has a background hard to see, Fig. 3 has low quality, Fig. 8 is upgradable, etc. (Check the Specific comments). Reply: we agree on the comments on the figures: in the revised version of the manuscript, we will reformat and reorganising them according to the suggestions from reviewers.

4. While the convergence line is a very important criteria for dynamics exploration, it hasn't been shown in any figure. Lines 273 and 274 signals the coordinates of this line but a graphical representation would clarify it. Reply: in the current version of the manuscript, the convergence lines corresponding to members 1, 13, 22 and 37 are highlighted by Figs. 10 and 11. These figures show the 10 m wind fields corresponding to the 4-hour periods with the minimum divergence values in Figure 9. In the revised version of the manuscript, we will highlight this point in the captions of figures 10 and 11, in order to better clarify that these figures correspond to the periods of minimum divergence in Figure 9.

Minor comments - L113 cites WRF version 2 while the work uses WRF version 3, the correct citation would be thus Skamarock et al. 2008 (NCAR/TN–475+STR) - L128 shows a good example in dates using sometimes upper-case and not using this. This is reproduced all over the paper. Consistency in the style should be shown. - L179 makes reference to Fig. 2b where it's shown 500hPa Geopotential but this is not stated in the text. Please add a comment on this field. - L280 text makes reference to QPF even if this abbreviation hasn't been introduced. Please define it. - L281 addresses

Fig. 13 while it should be Fig. 11. - L296 mentions a panel 6 which it's not shown in Fig. 10 Reply: we agree with all these comments and we thank the reviewer for these suggestions. The manuscript will be corrected accordingly.

––––––––––––––––––––––––––––––––––

[Figure]

[Figure]

**Fig. 1.** Member 1, 06UTC on 25 september 1915. Panel A shows the 2 m potential temperature field together with the 10 m horizontal wind vector field. Panel B to G show the vertical cross sections of potential

---

## Author Comment (AC3) · 4 Feb 2017

Now we have included figures 2 and 3.

Figure 2: QPF 1 km grid spacing 12UTC 24-09-1915 – 00UTC 26-09-1915 for the ensemble members 1, 13, 22 and 37: first row results corresponding to simulations driven by outermost domain at 25 km grid spacing, second row the same but with outermost domain at 125 km grid spacing.

Figure 3: Member 1, 06UTC on 25 september 1915. Panel A shows the 2 m potential temperature field together with the 10 m horizontal wind vector field. Panel B to G show

[Figure]

the vertical cross sections of potential temperature, vertical velocity, water vapour, rain water, snow, and graupel mixing ratios along the cross section corresponding to the green dotted line of Panel A.

[Figure]

[Figure]

**Fig. 1.** Figure 2.

[Figure]

**Fig. 2.** Figure 3.

---

## Author Response (AR1)

We thank the referees for their comments on the manuscript and for the many useful suggestions that helped us in preparing an improved version of the manuscript.

In the following, we address their comments.

**Referee #1**

**Main comments**

1. The 20th Century ensemble reanalysis at 2.0 degrees is used as boundary conditions for simulations with a 25km (~0.22 degree) outer domain. In the analysis, the poor alignment of the convergence zones is attributed to the low resolution of the reanalysis but this could be mitigated by introducing a WRF domain with 125km. This should be attempted at least for the four members analysed

**Reply: It is true that the ratio of the grid spacing for the driving data to that of our outer 25 km domain (roughly 1:9) is larger than what is normally recommended, and larger than the 1:5 ratio we use for our inner domains. Because of the large computational cost to run 56 ensemble members with each one including a refined 1 km domain, the decision originally had been made to only use 3 nests. Because we no longer have access to such a large amount of computer resources, we are unable to re-run all 56 members, but we recognize that the reviewer raises an interesting question. Therefore, with the limited resources remaining, we have chosen to re-run the 4 best members (1, 13, 22 and 37) adding a fourth outer domain with 125 km grid spacing (in addition to 25, 5, and 1 km grid spacing) (see Fig. 1).**

**The comparison of 36 hour QPF for the innermost domain at 1 km grid spacing is provided in Figure 2: first row results corresponding to simulations driven by outermost domain at 25 km grid spacing, second row the same but with outermost domain at 125 km grid spacing. Heavier areal QPF can be seen in all members in the first row (our original configuration), both on the entire 1 km grid spacing domain (both on sea and land areas), and also on the smaller area over which the paper focuses.**

**This statement is confirmed when comparing, for the 4 selected best members, the BIAS and MAE (in mm) over the available 64 raingauge stations in the runs using an outermost domain with 25 and 125 km grid spacing respectively (Table 1).**

| Member | BIAS-d01 25 km | BIAS-d01 125 km | MAE-d01 25 km | MAE-d01 125 km |
|--------|----------------|-----------------|---------------|----------------|
| 1 | -19.8 | -30.0 | 38.4 | 38.9 |
| 13 | -14.6 | -26.4 | 40.5 | 42.2 |
| 22 | -21.8 | -29.8 | 39.9 | 45.9 |
| 37 | -18.2 | -26.9 | 42.1 | 44.6 |

*Table 1. - BIAS and MAE (in mm) for the 4 selected best member ensembles over the available 64 raingauge stations in the runs having an outermost domain with 25 and 125 km grid spacing, respectively.*

The New Simplified Arakawa-Schubert (NSAS) scheme adopted in these additional simulations over the 125 and 25 km grid spacing domains has been revised, for deep moist convection, to make cumulus convection stronger and deeper to deplete more instability from the atmospheric column and result in the suppression of excessive grid-scale precipitation (Han and Pan, 2011). This can result, if applied even at very coarse grid spacing (125 km), in an overall reduction of the efficiency of the precipitation processes, thus impacting also the results on the innermost domains down to 1 km grid spacing.

We therefore believe it is advantageous to maintain our previously obtained results and to not introduce the extra 125 km domain, which adversely affects results, likely because of the NSAS scheme that we use.

2. The advantage of high resolution simulations is their ability to provide 3D information of an event. No analysis of the upper air results, vertical profiles or 2D vertical cross-sections has been presented. The dynamics of the storm evolution should be added to the manuscript.

Reply: the physical mechanism responsible for the generation of the back-building MCS observed on 25 september 1915 also has been recently explained by Fiori et al. (2017). Taking advantage of the availability of both observational data and modelling results at the micro-α meteorological scale, Fiori et al. (2017) provide insights about the triggering mechanism and the subsequent spatio-temporal evolution of the Genoa 2014 back-building MCS. The major finding is the important effect of a virtual mountain created on the Ligurian sea by the convergence of a cold and dry jet outflowing from the Po valley and a warm and moist low level south-easterly jet within the PBL.

The same mechanism is active also for this case. Let us consider, as an example, the convective flow field at 06UTC on 25 september 1915 (see Figure 3), as predicted by member 1 of the ensemble. Panel A shows the 2 m potential temperature field together with the 10 m horizontal wind vector field: the colder and drier jet outflowing from the Po valley and the warmer and moister air from southern mediterranean sea are evident. Panel B shows, by means of the potential temperature along the cross section corresponding to the green dotted line of Panel A, also the thin potential temperature layer (virtual mountain) in front of the actual Liguria topography. This acts, in agreement with Fiori et al. (2017), to produce the strong convective cells in panel C (updraft velocity above 10 m/s) with the apparent back-building on the western side (less mature and intense cells around 8.4° latitude). The main updraft produces vertical advection of water vapor (panel D), thus resulting in significant production of rainwater (panel E), snow (panel F, significantly advected inland by the upper level south-westerly winds), and graupel (panel G).

This analysis in now included in the revised version of the manuscript.

3. Although an ensemble of 56 members is produced, only 2 deterministic measures of individual ensemble members are presented but no analysis of the quality of such ensemble is provided. A shortcoming of deterministic measures of skill is that information about prediction uncertainties is not available, thus categorical measures like Brier skill score, continuous ranked probability score, ROC skill score are a useful tool to assess the quality of an ensemble forecast. In the following references examples such types of analysis can be found. Please add some categorical measures.

**Reply: It is well know (Mass et al. 2002) that point-to- point verification measures like those usually used for traditional ensemble verification do not work well with fine grid spacing simulations, because a double penalty exists for spatial errors, which are extremely common for high intensity precipitation events. This problem is likely even worse when limited observations from 1915 are used. Object-based verification techniques have been developed in the last 10-15 years specifically because of these problems. The application of the MODE Object-based verification technique showed that twelve members out of the 17 members selected using the minimum divergence criterion have significant values (above 0.8) of the total interest function. Specifically, when examining paired observed and modelled clusters, these twelve members demonstrate useful skill for: centroid distance, providing a quantitative sense of spatial displacement of forecast; forecast area/observed area, providing an objective measure of over-or under-prediction of areal extent of the forecasts; forecast intensity 50/observed intensity 50 and forecast intensity 90/observed intensity 90, providing objective measures of median (50th percentile) and near-peak (90th percentile) intensities found in the objects; and the already mentioned total interest, a summary statistic derived from the fuzzy logic engine with user-defined interest maps for all these attributes plus some others (Tab. 2).**

| Parameter | Average | Standard deviation |
|---|---|---|
| **PAIRED CENTROID DISTANCE (km)** | 114 | 62 |
| **FCST AREA/OBS AREA** | 1.10 | 0.90 |
| **FCST INT 50/OBS INT 50** | 0.73 | 0.06 |
| **FCST INT 90/OBS INT 90** | 0.62 | 0.11 |
| **TOTAL INTEREST** | 0.88 | 0.09 |

*Where:*
*CENTROID DISTANCE: provides a quantitative sense of spatial displacement of forecast.*
*FCST AREA/OBS AREA: provides an objective measure of whether there is an over-or under-prediction of areal extent of forecast.*
*FCST INT 50/OBS INT 50 and FCST INT 90/OBS INT 90 provide objective measures of Median (50th percentile) and near-Peak (90th percentile) intensities found in objects.*
*TOTAL INTEREST: provides summary statistic derived from fuzzy logic engine with user-defined interest maps for all these attributes plus some others.*

Table 2. - Clusters pairs statistics for the 12 members out of 17, showing significant values (above 0.8) of the total interest function.

**This results are now included and discussed in the revised version of the manuscript.**

4. The deterministic measures are also evaluated by comparing observations and simulations with different time spans. In lines 209-218 reference is made to rainfall depths for a 4 hour period thus QPE should be computed for the same time period as the simulation and only then should the evaluation be performed. In case that is not possible, the simulation should cover the same time period of the observations.

**Reply: A comparison at an hourly level is basically meaningless (due to high variability within the simulations) and also impossible (no observational data are available on**

**hourly scales). Furthermore the 12 hours covered by the observations and not covered by the model do not experience important precipitation (as supported with the notes on the past weather in the daily bulletins (e.g. "pioggia dal mattino" or "pioggia fino al pomeriggio"). Additionally the QPF in all 56 members for the period 12 UTC 24ᵗʰ – 00 UTC 25ᵗʰ is negligible over the entire Liguria Region (averaging below 1 mm in 12 hours). Therefore, verification statistics using the time periods we have chosen would not differ meaningfully from those performed if we had access to observations whose timing did match exactly the simulation period. We will mention this in the revised version of the manuscript.**

**Minor comments**

Figures 3 is very difficult to read. Since its quality cannot be enhanced I would suggest adding a figure 3b with the ensemble mean slp with the same domain and isobar resolution in order to better assess the resemblance between the 20th Century Reanalysis and the forecasted conditions on the 25th of September. The approximate pressure gradients in the Po Valley, Mediterranean and France, in both analysis, would be appreciated.

**Reply: Done.**

Figure 4c is as difficult to read as figure 3. Figures 4a and b should represent the same domain as figure 4c. Same argument as before**.**

**Reply: Done.**

In lines 209-218 reference is made to rainfall depths for a 4 hour period. If sub-daily precipitation is available please add either QPE or individual stations time series for the periods analysed in figures 10 and 11.

**Reply: Sub-daily precipitation is not available.**

The topography of the WRF plays a fundamental role in the development of the convective system but is missing from the manuscript. I suggest replacing the map in figure 8 with the model topography for all the domains.

**Reply: Done.**

To facilitate the comparison with the real topography, I would suggest the merger with figure 1 as figure 1b. Also in figure 1 there is no reference to the source of the topographic map.

**Reply: the prefer keeping figure 1 as it is, in order to avoid introducing in the introduction section a reference to technical issue that is presented only in section 3.**

Lines 95-100 – Paragraph is too long, please rephrase.

**Reply: Done.**

Line 170 – The paragraph refers to 500hPa chart, i.e. figure 2b Line 178 – Should be figure 2a, not 2b Line 746

**Reply: we corrected the mistake.**

Y axis in figure 9 difficult to read. Reduce the resolution and increase the caption font.

**Reply: We have added the measurement unit to the figure caption.**

Line 755 – Indexation of figure 10 and 11, hard to follow. Attribute the indices sequentially. Legend should describe better the individual panel figures.

**We improved the captions of figures 10 and 11. It was however not possible to attribute the indices sequentially because the 6 panels in figure 10 and the 6 ones in figure 11 refer to different ensemble members.**

**Referee #2**

**Main comments**

1. Even if 56 members on the 20th century reanalysis were studied, only four of them reproducing the best the event's dynamics were taken into account while showing the results. It would be interesting to have some comments about the members showing very "non-realistic dynamics" and also about the mean ensemble.

**Reply: some information on the other members would indeed be interesting, but we prefer keeping the focus of the paper which investigates the ability of the ARF-WRF simulations to capture the MCS character of the event. The members showing very "non-realistic dynamics" and also the mean ensemble fail to capture the convergence line creation and its evolution responsible for the generation of the back-building MCS in 17 out of 56 members. We prefer therefore not discussing them in order to keep the focus of the paper on the back-building MCS character of the investigated event.**

2. Convective systems are generally associated with vertical motion. WRF outputs offers 3D information allowing the generation of vertical cross-section plots or Skew-T diagrams, none of them are shown in the paper. Some graphs and words about this should be added.

**Reply: the physical mechanism responsible for the generation of the back-building MCS observed on 25 september 1915 also has been recently explained by Fiori et al. (2017). Taking advantage of the availability of both observational data and modelling results at the micro-α meteorological scale, Fiori et al. (2017) provide insights about the triggering mechanism and the subsequent spatio-temporal evolution of the Genoa 2014 back-building MCS. The major finding is the important effect of a virtual mountain created on the Ligurian sea by the convergence of a cold and dry jet outflowing from the Po valley and a warm and moist low level south-easterly jet within the PBL.**

**The same mechanism is active also for this case. Let us consider, as an example, the convective flow field at 06UTC on 25 september 1915 (see Figure 3), as predicted by member 1 of the ensemble. Panel A shows the 2 m potential temperature field together with the 10 m horizontal wind vector field: the colder and drier jet outflowing from the Po valley and the warmer and moister air from southern mediterranean sea are evident. Panel B shows, by means of the potential temperature along the cross section corresponding to the green dotted line of Panel A, also the thin potential temperature layer (virtual mountain) in front of the actual Liguria topography. This acts, in agreement with Fiori et al. (2017), to produce the strong convective cells in panel C (updraft velocity above 10 m/s) with the apparent back-building on the western side (less mature and intense cells around 8.4° latitude). The main updraft produces vertical advection of water vapor (panel D), thus resulting in significant production of rainwater (panel E), snow (panel F, significantly advected inland by the upper level south-westerly winds), and graupel (panel G).**

**This analysis in now included in the revised version of the manuscript.**

3. In general the writing style and content is of good quality but the graphs are not at the same level of quality. Fig. 2 has a background hard to see, Fig. 3 has low quality, Fig. 8 is upgradable, etc. (Check the Specific comments).

**Reply: we agree on the comments on the figures: the figures of the revised version of the manuscript have been reformat and reorganising according to the suggestions from reviewers.**

4. While the convergence line is a very important criteria for dynamics exploration, it hasn't been shown in any figure. Lines 273 and 274 signals the coordinates of this line but a graphical representation would clarify it.

**Reply: in the current version of the manuscript, the convergence lines corresponding to members 1, 13, 22 and 37 are highlighted by Figs. 10 and 11. These figures show the 10 m wind fields corresponding to the 4-hour periods with the minimum divergence values in Figure 9. In the revised version of the manuscript, we.**

**Minor comments**

- L113 cites WRF version 2 while the work uses WRF version 3, the correct citation would be thus Skamarock et al. 2008 (NCAR/TN–475+STR)

**Reply: we corrected the reference.**

- L128 shows a good example in dates using sometimes upper-case and not using this. This is reproduced all over the paper. Consistency in the style should be shown.

**Reply: Done; all dates are now upper-case.**

- L179 makes reference to Fig. 2b where it's shown 500hPa Geopotential but this is not stated in the text. Please add a comment on this field.

**Reply: we corrected the mistake.**

- L280 text makes reference to QPF even if this abbreviation hasn't been introduced. Please define it.

**Reply: Done.**

- L281 addresses Fig. 13 while it should be Fig. 11.

**Reply: we corrected the mistake.**

- L296 mentions a panel 6 which it's not shown in Fig. 10

**Reply: we corrected the mistake.**

[Figure]

Figure 1: ARF-WRF 4-domains setup.

[Figure]

**Figure 1: QPF 1 km grid spacing 12UTC 24-09-1915 – 00UTC 26-09-1915 for the ensemble members 1, 13, 22 and 37: first row results corresponding to simulations driven by outermost domain at 25 km grid spacing, second row the same but with outermost domain at 125 km grid spacing.**

[revised manuscript text omitted]

---

## Author Response (AR2)

We thank the referees for their comments on the manuscript and for the many useful suggestions that helped us in preparing an improved version of the manuscript.

In the following, we address their comments.

**Main comment 1 of referee #2:**

I agree that the focus should be on those members that work well. However, I still think that the others simulations should be mentioned from time to time. For instance, the abstract just says "those simulations" without giving a number. At this point it would be interesting for the reader whether this was 1 or 20 members. I do not think it needs a lot of detailed analyses, but it is important to keep that selection process always in mind.

**Reply: agreed. Two senteces have been added to the new manuscript version, see line 27 and lines 270-272.**

**Main comment 3 of referee #2:**

The figures in the revised version are still of varying quality. Those figures (Fig. 2 to 4) that were taken directly from the NOAA/ESRL plotting tools website (Figs 2-4) contrast with the other ones. I fear that coast lines will hardly be recognizable in print, and the same might hold for the contour labels.

**Reply: agreed. Figures 2, 3, 4 have been redrawn.**